# Diet, Sun, Physical Activity and Vitamin D Status in Children with Inflammatory Bowel Disease

**DOI:** 10.3390/nu14051029

**Published:** 2022-02-28

**Authors:** Karolina Śledzińska, Piotr Landowski, Michał A. Żmijewski, Barbara Kamińska, Konrad Kowalski, Anna Liberek

**Affiliations:** 1Department of Internal and Pediatric Nursing, Faculty of Health Sciences with Institute of Maritime and Tropical Medicine, Medical University of Gdansk, 80-211 Gdansk, Poland; anna.liberek@gumed.edu.pl; 2Department of Pediatrics, Pediatric Gastroenterology, Allergology and Nutrition, Medical University of Gdansk, 80-803 Gdansk, Poland; piotr.landowski@gumed.edu.pl (P.L.); bkam@gumed.edu.pl (B.K.); 3Department of Histology, Medical University of Gdansk, 80-211 Gdansk, Poland; 4Masdiag Sp. z o.o. Company, 01-882 Warsaw, Poland; konrad.kowalski@masdiag.pl

**Keywords:** vitamin D, inflammatory bowel disease, children, diet, sun

## Abstract

In the course of inflammatory bowel disease (IBD) malabsorption may lead to a vitamin D deficiency and calcium–phosphate misbalance. However, the reports on the vitamin D status in children with IBD are few and ambiguous. Here, we are presenting complex analyses of multiple factors influencing 25OHD levels in IBD children (*N* = 62; Crohn’s disease *n* = 34, ulcerative colitis *n* = 28, mean age 14.4 ± 3.01 years, F/M 23/39) and controls (*n* = 47, mean age 13.97 ± 2.57, F/M 23/24). Additionally, calcium–phosphate balance parameters and inflammatory markers were obtained. In children with IBD disease, activity and location were defined. Information about therapy, presence of fractures and abdominal surgery were obtained from medical records. All subjects were surveyed on the frequency and extent of exposure to sunlight (forearms, partially legs for at least 30 min a day), physical activity (at least 30 min a day) and diet (3 days diary was analyzed with the program DIETA 5). The mean 25OHD level was higher in IBD patients compared to controls (18.1 ng/mL vs. 15.5 ng/mL; *p* = 0.03). Only 9.7% of IBD patients and 4.25% of controls had the optimal vitamin D level (30–50 ng/mL). Despite the higher level of 25OHD, young IBD patients showed lower calcium levels in comparison to healthy controls. There was no correlation between the vitamin D level and disease activity or location of gastrointestinal tract lesions. Steroid therapy didn’t have much influence on the vitamin D level while vitamin D was supplemented. Regular sun exposure was significantly more common in the control group compared to the IBD group. We found the highest concentration of vitamin D (24.55 ng/mL) with daily sun exposure. There was no significant correlation between the vitamin D level and frequency of physical activity. The analysis of dietary diaries showed low daily intake of vitamin D in both the IBD and the control group (79.63 vs. 85.14 IU/day). Pediatric patients, both IBD and healthy individuals, require regular monitoring of serum vitamin D level and its adequate supplementation.

## 1. Introduction

The contribution of vitamin D in the regulation of the calcium–phosphate balance and its metabolic complications (rickets, osteoporosis, osteopenia) is widely recognized. Recent studies have highlighted the possible role of vitamin D in the pathogenesis of numerous diseases, including autoimmune, endocrine, metabolic, cardiovascular and gastrointestinal diseases and cancer [1,2,3,4]. It is well established that skin production of vitamin D, induced by sun exposure, is its best source. To achieve the optimal serum 25OHD concentration, residents of Central Europe should expose approximately 18% of the body surface area (e.g., forearms and partially leg) without protective sun blockers, 2–3 times a week, for a minimum of 15 min, between 10 a.m. and 3 p.m., from April to September [2]. However, ultraviolet B fraction of the sunlight, which is essential for the photo conversion of 7-dehydrocholesterol to vitamin D, is also a known carcinogen. Thus, limited and controlled exposition to the sun (main source of UVB) is advised [2,5,6].

On the other hand, heterogeneous diet rich in oily fish (mackerel, salmon, sardines) and eggs may cover the daily intake of vitamin D, but usually it is very difficult to compose such a vitamin D-rich diet every day. Having in mind potential risks and the overall indoor lifestyle of modern societies, oral supplementation is needed in order to prevent a vitamin D deficiency.

Any anatomical disorders or dysfunction of the gastrointestinal tract may impact vitamin D and calcium absorption and influence its proper functions [7]. One of the examples is malabsorption in the course of inflammatory bowel disease (IBD), that potentially may lead to a calcium–phosphorus misbalance.

There is growing evidence of the pleiotropic effect of vitamin D, including its influence on immune-mediated diseases, such as IBD where chronic dysbiosis may cause a vitamin D dysfunction [8,9]. Vitamin D is an important regulator of immune response, intestinal immunity and calcium absorption in the small intestine by the stimulation of the expression of several transepithelial Ca^2+^-transporter genes, epithelial barrier renewal and its regeneration [10,11,12,13]. However, not much is known concerning vitamin D’s status in IBD pediatric patients [11,14,15]. One may presume that children with IBD have reduced sun exposure due to recurrent hospitalizations or the type of therapy where sun exposure is contraindicated. Additionally, their diet may be poor in milk products (because of frequent lactose intolerance) or other food rich in vitamin D because of typical symptoms of the disease (diarrhea, abdominal pain) that may occur afterwards [9,14,16,17].

The aim of the study was to assess the extent of a possible vitamin D deficiency in children with IBD compared to controls and to determine factors that may affect the vitamin D level. In addition, we considered indications for vitamin D supplementation in both study groups. This represents novel and complex analyses of factors affecting the vitamin D status in children with IBD.

## 2. Materials and Methods

### 2.1. Patients and Controls

The study involved 109 individuals, including 62 children diagnosed with IBD and 47 children defined as control/comparative group (C). Table 1 shows the detailed characteristics of study groups. Mean age of IBD patients was 14.4 ± 3.01 years, and of the controls—13.97 ± 2.57. All individuals had their height and weight measured and BMI calculated with corresponding z-scores values added (Table 2). The criterion for qualifying to study group was the diagnosis of IBD, which was based on clinical presentation, endoscopic and histopathological assessment of colon mucosa and laboratory test results (PORTO criteria) [14,16,17]. All patients with IBD were evaluated for disease activity using PCDAI/PUCAI score [15,16,18,19]. Disease location for Crohn’s disease (CD) was described as L1 (distal 1/3 ileum +/− cecal disease), L2 (colonic), L3 (ileocolonic) and L4 (with upper gastrointestinal tract involvement). Ulcerative colitis (UC) was defined as E1 (proctitis), E2 (left sided—distal to splenic flexure), E3 (extensive—hepatic flexure distally) and E4 (pancolitis—hepatic flexure proximally) [17,20]. Young patients with any inflammatory, infectious, nutritional, immunological and cancer disorders 6 months prior to enrollment were excluded from the control group.

### 2.2. Past Medical History

Patients’ data were retrieved from medical records, including: duration of the disease, types of therapy used, potential presence of fractures and abdominal surgery due to complications of the underlying disease. All individuals from IBD and control group were asked questions regarding the frequency of sun exposure of arms and legs for at least 30 min a day at different times of the year. A point scale was used to assess the frequency of sun exposure of IBD and control group patients—daily sun exposure was given 3 points, when the exposure took place several times a week—2 points, few times a month—1 point, less often—0 points, regardless of the season. A maximum of 12 points could be obtained (4 seasons with a maximum number of 3 points; 4 × 3 = 12), a minimum of 0 points. Similarly, all individuals from IBD and control group were asked questions regarding the frequency of at least 30 min of physical activity at different times of the year. A point scale was used to assess the frequency of physical activity of IBD and control group patients—30 min per day daily activity was given 3 points, when the activity took place several times a week—2 points, few times a month—1 point, less often—0 points, regardless of the season. A maximum of 12 points could be obtained (4 seasons with a maximum number of 3 points; 4 × 3 = 12), a minimum of 0 points.

Patients’ diet was analyzed as a questionnaire regarding the frequency of consumption of products rich in calcium and vitamin D (fatty fish, eggs, milk, yellow cheese, yogurt) plus 3-days’ diary was analyzed with program DIETA 5 (National Institute of Public Health—National Institute of Hygiene, Warsaw, Poland).

All studies were conducted with the written consent of parents/legal guardians of children and patients who were 16 years of age. The above-mentioned research was approved by the Independent Bioethics Commission for Research at Gdansk Medical University No. 272/2011.

### 2.3. Measurements of 25OHD

The concentration of 25OHD was determined by the immunochemiluminescence method using the DiaSorin Liaison XL analyzer (DiaSorin S.p.A., Saluggia VC, Italy; Chemiluminescent immunoassay (CLIA)). Samples were furtherly realized for 25OHD quantification using liquid chromatography coupled with tandem mass spectrometry (QTRAP^®^4500 (Sciex, Framingham, MA, USA) coupled with ExionLC HPLC system (Sciex, Framingham, MA, USA); Liquid Chromatography – Mass Spectrometry (LC-MS) method), as published previously [21].

### 2.4. Statistics

Initial comparison of CLIA and LC-MS quantification of the 25OHD concentrations showed convergence of the results, with slightly higher readings for LC-MS. Furthermore, preliminary comparisons between the groups (CD vs. C, CU vs. C) showed the same relationships, therefore, further statistical analysis was performed on the data obtained by the CLIA method.

Data was collected and pre-prepared in Excel spreadsheet (Microsoft Office 2010). STATISTICA 8.0 software (StatSoft Polska, Kraków, Poland) was used for statistical calculations. Basic characteristics of quantitative variables have been established—arithmetic mean, median, minimum, maximum and standard deviation. Each quantitative variable was subjected to Shapiro–Wilk’s normality analysis. Differences were statistically significant at *p* < 0.05 (*t*-Student test or Mann–Whitney–Wilcoxon test). The degree of mutual interdependence was determined by Pearson or Spearman correlation coefficients. Qualitative variables were sorted and analyzed using the chi-squared test with or without Yates correction (by sample size).

## 3. Results

### 3.1. Calcium-Phosphate Metabolism Parameters and Inflammatory Markers Status in IBD Pediatric Patients

Median 25OHD concentration in the IBD group was 18.1 ng/mL and was significantly higher than in controls (15.5 ng/mL) (*p* = 0.03) (Figure 1). Vitamin D deficiency (25OHD < 20 ng/mL) was identified in the majority of IBD patients (59.7%) and comparative group (80.85%) (Figure 2). The level of vitamin D was lower in girls (17.25 ng/mL) compared to boys (18.95 ng/mL) in the total group of patients (n = 109), but the result was not statistically significant (*p* = 0.32). There was no significant correlation between the vitamin D level and age of patients (R = −0.004, *p* = 0.966). Surprisingly, the highest vitamin D level was seen in IBD patients during the winter; however, this observation could be explained by vitamin D supplementation (Table 3). Analysis of laboratory results (blood taken simultaneously as for 25OHD analysis), revealed that in the IBD group, inflammatory markers were significantly higher compared to controls (CRP *p* = 0.04, WBC *p* = 0.04, neutrophils *p* = 0.03, PLT *p* < 0.001). In addition, in the IBD group, the hemoglobin level was significantly lower compared to controls (*p* < 0.05). Among calcium–phosphate balance parameters, only the calcium level was significantly lower in IBD patients vs. controls, the other did not differ significantly between analyzed groups (PTH, P, ALP) (Table 4). Interestingly, there was no correlation found between the vitamin D and PTH level (*p* > 0.05), nor the calcium and PTH level.

### 3.2. Clinical Course of the Disease and Vitamin D Level

Based on the PCDAI/PUCAI score, at the time of the study most children with CD had a mild form of the disease (47.06%) or no disease activity (29.41%); the majority of patients with UC (46.43%) had remission or mild activity and there were no patients in a severe condition (Table 1). The highest concentration of 25OHD was found in patients with CD with a moderate form of the disease (25.9; 20.25–41.5 ng/mL) and the lowest in UC patients with severe intensity (13.47 ± 1.58 ng/mL) (Appendix A). However, differences were not statistically significant. There was no significant difference in 25OHD level regarding the location of the disease in CD (L1/L2/L3) or UC (E1/E2/E3/E4). The mean duration of illness in patients with CD was 29 months and in children with UC it was 16 months. No correlation was found between the vitamin D concentration and the duration of the disease (Spearman coefficient = 0.038, *p* > 0.05). In addition, no correlation was found between the presence of abdominal surgical interventions and the vitamin D level in IBD patients. In patients with IBD, fractures occurred more frequently in comparison to the controls (17.74% vs. 6.38%, *p* = 0.079). In addition, within the IBD group children with CD had the highest prevalence of fractures (20.59% CD vs. 14.29% UC vs. 6.38% control group; *p* = 0.16). However, there was no statistically significant correlation between the 25OHD level in IBD patients and the history of bone fractures (*p* = 0.37).

### 3.3. Vitamin D Level and Nutritional Status of ID Children

Children with IBD had a significantly lower weight z = score compared to controls (IBD/CD/UC vs. C *p* < 0.001) and children with IBD and CD had a significantly lower BMI z-score than controls (IBD/CD vs. controls *p* = 0.008) (Table 2). There was no correlation between the vitamin D level and nutritional status expressed as weight/height/BMI z-score in the IBD/CD/UC and control groups. Furthermore, there was no significant difference in IBD patients in anthropometric values in two subgroups of patients: those with a vitamin D deficiency and those without (Appendix A). We observed that the higher the BMI z-score, the higher the vitamin D level, but the difference wasn’t significant (Appendix A).

### 3.4. Sun Exposure

The frequency of sun exposure between groups was significantly different in spring, autumn and winter seasons. Regular sun exposure (i.e., daily or several times a week) was reported significantly more frequently in the control group compared to the study group. The difference was the smallest in the summer (Table 5).

There was no significant difference in the number of points obtained between the IBD group and control group (IBD—8.0 (6.0–11.0); C—8.0 (8.0–9.75), *p* > 0.05). Next, the vitamin D concentration was determined in patient subgroups depending on the number of points obtained. There is a visible pattern of an increasing vitamin D level with the increasing number of points (more frequent sun exposure throughout the year) with the significantly highest vitamin D concentration (24.55 ng/mL) in patients exposed to sun daily (Table 6). Similar observations were found when the total group was monitored only during summer season (Table 7). Moreover, when the IBD and control group were analyzed separately, the similar tendency was confirmed (Table 8).

The increasing tendency of vitamin D level with declared increasing frequency of sun exposure was shown in both the supplemented and non-supplemented IBD group with the higher vitamin D level in the supplemented group of patients (Table 9 and Table 10).

### 3.5. Physical Activity

The control group declared every day and few times a week for physical activity, more often than the IBD group, in every season of the year, but the difference wasn’t significant (Appendix A).

There was no significant difference in the number of points obtained between the IBD group and control group (IBD—8.0 (8.0–12.0); C—9.0 (8.0–12.0), *p* > 0.05). Next, the vitamin D concentration was determined in subgroups of patients depending on the number of points obtained (Appendix A). There was no significant difference in the level of vitamin D and declared frequency of physical activity in the total group of patients.

### 3.6. Vitamin D and Diet

Milk, yogurt and yellow cheese were consumed more frequently by the control group (Table 11). Moreover, a three days’ dietary diary was collected from all of the individuals. Significant differences were found in the % EAR (estimated average requirement) and in the amount of consumed calcium (Table 12). There was no correlation between the level of vitamin D and amount of vitamin D consumed calculated from the dietary diary (Spearman—0.102, *p* = 0.428).

### 3.7. Medications and Vitamin D Level

At the time of the study, 29 (46.7%) patients with IBD were on steroid therapy, 33 (53%) patients were taking vitamin D supplements, however, only 17 patients (27%) were simultaneously on steroid therapy and vitamin D supplementation (Table 1). In the control group, only 7 people (14.9%) took vitamin D supplements. Overall, supplementation with vitamin D had a significant effect on the 25OHD level for all children in the study (IBD and control) and resulted in an increase from 14.85 ng/mL (12.45–19.4) to 19.8 ng/mL (14.1–25.2) in the supplementing group (*p* = 0.005). The daily dose of vitamin D in all study groups varied from 200 IU to 6000 IU. Interestingly, the significantly higher concentration of vitamin D was observed with vitamin D supplementation of 601–1000 IU/day in CD and IBD patients (Table 13).

Steroid treatment didn’t influence the level of vitamin D as much as vitamin D supplementation; the vitamin D concentration in the IBD and CD group supplementing vitamin D regardless of the steroid treatment and time of the year, was higher compared to those not on vitamin D (Table 13).

There was no statistically significant difference in the vitamin D level between the group taking immunosuppressants compared to the non-treatment group (Table 13) (*p* > 0.05). Vitamin D levels in IBD children who were on biological treatment were higher (26.6 ng/mL) compared to others; the difference wasn’t significant. Patients with IBD receiving nutritional treatment had a higher vitamin D level, but the difference wasn’t statistically significant (Table 13).

## 4. Discussion

### 4.1. Laboratory Test Results

Vitamin D status analysis among IBD patients has been a subject of numerous publications, but only few referred to the pediatric population. Moreover, the results are still inconclusive; the vitamin D level in IBD patients usually is lower than in the control group or there is no significant difference [15,22,23,24,25,26,27,28,29]. To our knowledge, this is the first study that considers multiple variables that may influence the vitamin D level in IBD children.

In our study, the median 25OHD concentration in the IBD group, contrary to our initial hypothesis, was significantly higher than in the control group (18.1 ng/mL vs. 15.5 ng/mL) (*p* = 0.03). However, almost 60% of IBD patients and 80% of the control group had a vitamin D deficiency defined as 25OHD < 20 ng/mL. This could be at least partially explained by the significantly higher number of IBD patients taking vitamin D supplements (53%) in comparison to the control group (14%). The increased frequency of supplementation is, in turn, probably caused by greater healthcare awareness of IBD families due to recurrent clinic/hospital visits. It is worth to mention that only 13.9% of IBD patients and 4.25% children in the control group have a sufficient level of vitamin D (above 30 ng/mL) according to recent guidance [1,2]. Thus, adequate supplementation is advised for both groups.

In our study, the characteristics of our IBD study group are comparable to epidemiological IBD data described in the literature [30]. Most investigators, similarly to our study results, do not show any correlation between the vitamin D level and age or gender of subjects in both the IBD and the control group [31]. In our study, we have shown that weight and BMI z-score were significantly lower in children with IBD.

Malnutrition and related delays in growth and puberty are one of the most common complications of IBD in children, but may also be one of the first symptoms of the disease. Its pathogenesis is multifactorial, and causes include chronic malnutrition, as well as the effect of pro-inflammatory cytokines released by the intestinal inflammatory process, which in turn has an adverse effect on bone metabolism [8,10,28]. On the other hand, although there is no correlation between 25OHD and BMI z-score in children with IBD in our study, the literature has highlighted the association of overweight or obesity (expressed as BMI z-score) with vitamin D deficiency in both healthy and non-healthy children [15,32,33]. One of the main hypotheses as a cause of vitamin D deficiency indicates the sequestration of this compound in adipose tissue, which lowers its bioavailability [2,32].

Abnormalities found in the laboratory blood results of our IBD patients (CRP, white blood cell count and platelet count, granulocyte count) are consistent with PORTO criteria, where evidence of decreased hemoglobin, elevated inflammatory markers, elevated platelet counts and low plasma albumin levels in patients presenting symptoms of IBD strongly predicts the diagnosis of IBD [17]. On the other hand, it was shown that supplementation with high doses of vitamin D (50,000 IU) can modify markers of inflammation including CRP and neutrophil to the lymphocytes ratio [34]. Further analysis revealed only significantly lower calcium level in IBD patients vs. controls, other parameters did not differ significantly between the analyzed groups (PTH, P, ALP).

### 4.2. Clinical Course of the Disease and Vitamin D Level

Analysis of the vitamin D level and disease activity in the present study did not show simple dependence. The literature on this subject is also unclear [35,36]. El-Matary analysis of vitamin D concentration in 60 children with newly diagnosed IBD did not show any association with the severity of the disease, while Joseph et al., in 34 adult Crohn’s patients, showed a negative correlation [25,37]. There was no correlation between the duration of the disease and vitamin D level found in our study. Similarly, Levin et al., analyzing 78 children with IBD, reported a lower vitamin D level (70.7 nmol/L, 28.28 ng/mL) in the group of children with a longer lasting illness (more than 70 days), but the difference was not statistically significant (*p* = 0.2) [31].

We didn’t find any correlation between the vitamin D level and location of the endoscopic lesions as well. Since in CD patients, patients with extensive endoscopic lesions showed higher serum vitamin D levels compared to those with limited colonic lesions, we might presume of less severe absorption abnormalities in the course of the disease than previously thought. Data from studies analyzing this association in IBD children require further studies due to the small sample size [31], although some researchers suggest a higher prevalence of vitamin D deficiency in patients with Crohn’s disease with upper gastrointestinal tract involvement [32,38]. In the literature it is suggested that resection of the small intestine may be a risk factor for vitamin D deficiency [39,40]. In our study, we did not confirm that hypothesis, and other researchers—mainly in pediatric studies—did not show such a correlation as well [41].

In the analyzed group of children, bone fractures were more common in patients with IBD (17.78%; CD-7 vs. UC-4) than in the control group (6.38%), with no correlation to the vitamin D level, probably due to small sample size.

Disorders of mineralization and bone metabolism appear in children with IBD pretty often as a result of many factors. It seems that one of the most important is the expression of proinflammatory cytokines which have a negative effect on osteoblast function and differentiation, bone formation and matrix mineralization. Adverse effects are also seen in prolonged high dose steroids therapy, reduced lean body mass, inadequate nutrition, vitamin D deficiency, calcium deficiency, growth disorders and delayed maturation associated with relative hormonal deficiency (gonadal steroids) or resistance (growth hormone and IGF-1). Data from the literature suggest that inflammatory bowel disease does not affect the incidence of long bone fractures, but may be a risk factor for vertebral fractures [42,43]. As it has been described that children with IBD have a decreased bone mineral density as compared to their healthy peers, it would be interesting to perform further analyses of the correlation of the vitamin D level and BMD [41,43].

### 4.3. Sun Exposure

In present study, at the time of the analysis, the majority of the research group and the entire comparative group had laboratory tests (including 25OHD level) done in the winter season. Among the studied groups, the highest vitamin D concentration was described in the IBD group tested during the winter, but it could be explained by vitamin D supplementation (Table 2). Therefore, we analyzed the sun exposure factor more thoroughly. Most IBD children declared that they were staying at least 30 min outdoors a day every day or several times a week throughout the year, but children in the comparison group spent more time outdoors altogether. In the analyzed group of children, the vitamin D concentration correlated with the frequency of sun exposure. This finding is in accordance to the previous study [44] where summer sun exposure was found to be a major contributor to the 25OHD level in the study group. It is consistent with other studies suggesting that one of the causes of vitamin D deficiency in IBD patients may be reduced sun exposure due to the fact of recurrent hospitalization, decreased time spent outdoors, and avoiding sun because of immunosuppressive and steroid therapy [14,31,41]. Reduced exposure to sunlight and consequent reduced vitamin D skin synthesis resulting in its deficiency have been the subject of numerous studies analyzing the etiology of many autoimmune diseases, which are more prevalent in the northern latitudes. Several studies indicated lower levels of vitamin D in IBD patients during the autumn and winter seasons, which could be explained by less sun exposure [31,32,45,46]. Another analysis of 623 pediatric IBD patients revealed no association of IBD onset with the seasonal pattern, whereas flares were more common in June and less common in April, with a significantly lower vitamin D level during flares compared with remission [47]. Limketkai et al., in retrospective analysis performed on an enormous population of over one million people, IBD patients and controls (649,932 patients with CD, 384,267 patients with UC and 288,894 patients in the control group) showed that reduced exposure to sun was associated with a higher risk of hospitalization, longer hospitalization, and more frequent surgical interventions [48].

### 4.4. Vitamin D and Diet

In order to ensure an adequate vitamin D supply from diet one should consume a dozen eggs a day, a large amount of fatty fish or fortified dairy products or cereal [2]. Composition of such a menu is practically impossible. NASPGHAN (North American Society for Pediatric Gastroenterology, Hepatology and Nutrition) Guidelines for Care of Children and Young People with IBD recommends a routine evaluation of their nutrition, with particular attention to the intake of vitamin D-rich products [49]. The analysis of the dietary diaries of the study group showed a low average daily intake of vitamin D and calcium in both the IBD and the control group. The calculated Ca:P ratio was abnormal in IBD patients—0.59 (CD 0.61, UC 0.57), and in the comparison group 0.6 (normal value for Ca:P at this age group is 1.32) which is consistent with recently published data [50].

IBD patients often avoid dairy products because they are afraid of symptoms of lactose intolerance [39]. Consequently, in addition to the low dose of vitamin D, they do not provide the body with the right amount of calcium. Currently, it is advisable to avoid dairy products in patients with IBD only if undesirable symptoms such as diarrhea, bloating and abdominal pain appear after consumption. It is not recommended to avoid milk products in advance [51]. There are only few studies analyzing IBD patients’ diets. Shoda et al., in a study conducted in Japan in 1966–1985, showed that the increased risk of CD was correlated with a higher consumption of fat, n-6 polyunsaturated fatty acids and dietary protein [52]. A literature review by Hou et al. analyzes the diets of patients with IBD prior to onset and showed an increased refined sugar intake in patients with CD [53]. However, intervention studies using a diet rich in fiber and low in sugars did not produce positive results. The potential cause could also be excess of insoluble fiber in the diet [53,54].

### 4.5. Physical Activity

Physical activity plays an important role in the normal development of children and adolescents. Daily physical activity positively affects respiratory and circulatory capacity, muscle strength, stimulates normal growth and bone strength, and has a positive effect on cognitive function and psychosocial factors. Low physical activity is a risk factor for cardiovascular disease, diabetes or osteoporosis [55,56,57]. In the analysis, 81.59% of children with CD and 81.24% of UC declared physical activity daily or several times a week for at least 30 min throughout the year compared to 94.15% of children in the comparative group. Physical activity was most often practiced in the spring. Studies on the assessment of physical activity in children with IBD are few. In a study by Werkstetter et al. it was demonstrated that children and adolescents, despite a well-controlled course of illness, showed reduced physical activity, especially girls with mild illness [55]. Physical activity during remission or in the phase of mild disease activity is not contraindicated, but IBD patients are generally reluctant to exercise despite their relatively good health, and in the periods of exacerbation, physical activity is even more limited.

### 4.6. Medications and Vitamin D Level

In the current study, 53% of IBD patients supplemented vitamin D. There are no consistent guidelines regarding optimal vitamin D supplementation or treatment of a deficiency in IBD pediatric patients. Ideally, it should maintain 25OHD concentration above 30–32 ng/mL to achieve optimal intestinal calcium absorption and to reduce PTH activity [2,58]. British guidelines advise, especially in the elderly or those on steroid therapy, a daily dose of vitamin D of 700–800 IU and 500–1000 mg of calcium, that could reduce the risk of hip and non-vertebral fractures [59]. Interesting meta-analysis of vitamin D treatment regimes in IBD children revealed no clear conclusion [29,32]. Wingate et al. showed a better effect of 2000 IU vs. 400 IU, which led to a vitamin D sufficiency (30 ng/mL) in 74% of CD subjects after 6 months of supplementation [36]. On the other hand, Pappa et al. revealed better results with vitamin D doses of 50,000 IU weekly vs. 2000 IU daily [32]. Increasing the dose even higher, Shepherd et al. presented a treatment with a single high age-adjusted dose of vitamin D (stoss therapy) that kept the vitamin D level above 32 ng/mL in 98% of the subjects at 1 month, 63% of the subjects at 3 months and at 6 months, 17% of the subjects had not maintained 25OHD concentration of 20 ng/mL or above. No significant adverse effects were noted [35]. Pludowski et al. has suggested that supplementation up to 10,000 IU daily is relatively safe and could be used in the treatment of severe vitamin D deficiency [1,2]. Lower doses of vitamin D maintained vitamin D sufficiency for a shorter period of time [28]. So far, there is not enough evidence to confirm the benefits of vitamin D supplementation in the treatment of IBD. There are some studies, the majority in adults, evaluating the influences of a vitamin D deficiency at the course and severity of the disease, but the results are conflicting [8,29,35,36,39]. It could be postulated that doses of 600 IU–1000 IU of vitamin D (used in this study) are not sufficient for IBD pediatric patients, which would explain the high prevalence of vitamin D deficiencies in IBD children despite supplementation.

The treatment used in the analyzed group of children was consistent with ECCO/ESPGHAN recommendations [60,61]. Experts have listed a group of children with UC as a risk group for vitamin D deficiency, although no clear association with osteopenia has been demonstrated. Regular monitoring of serum 25OHD levels and appropriate supplementation based on serum 25OHD levels are recommended, especially in patients with a reduced bone density [60,61]. Levin et al. demonstrated that children with IBD, who received higher doses of steroids had a lower vitamin D level as compared with those who received lower doses of steroids [31]. There is no direct relationship between the activity of the disease or its location and the level of vitamin D. However, with the implementation of steroid therapy as a measure of the severity of IBD, it can be indirectly deduced that patients on steroid therapy had more severe lesions. Perhaps the lower vitamin D levels in these patients are not only the result of steroid therapy, but also the severity of inflammatory changes [31]. The ECCO/ESPGHAN recommendations for the treatment of children with IBD emphasize the importance of nutritional treatment, especially in patients with CD [62]. In the present study, the vitamin D concentration in children on nutritional treatment was higher (19.5 ng/mL) vs. the untreated group (17.85 ng/mL) (*p* > 0.05), which could be explained by the increased intestinal absorption due to “mucosal healing” initiated by nutritional therapy. It is also known that nutritional treatment has a positive effect on bone turnover [63].

The higher vitamin D level in IBD children (22.6 ng/mL vs. 17.45 ng/mL in not treated group; *p* > 0.05) treated with biological drugs, such as infiximab, can be explained by known the tumor necrosis factor (TNF alpha) effect on bone metabolism. It increases the expression of RANK-L, an osteoclast differentiation factor (ODF), leading to the formation of osteoclasts and stimulation of bone resorption [64]. The REACH study on infliximab in the induction and supportive therapy in children with CD showed an impressive increase in bone formation markers after 10 weeks of treatment [65].

The results obtained in this study and the literature review may be the basis for further analysis. More and more reports underline the role of pleiotropic action of vitamin D, but there is still a lack of properly designed randomized trials for patients with IBD, especially in the pediatric population. Although IBD patients had a higher vitamin D level than controls, which probably resulted from more frequent supplementation, still, 60% of IBD patients were diagnosed with a vitamin D deficiency. Therefore, it is important to emphasize the necessity of vitamin D supplementation in both chronically affected and healthy individuals. It is important to introduce vitamin D supplementation to all pediatric patients, especially during the winter season or those on steroid therapy. Although the major negative effect of vitamin D deficiency is its influence on bone metabolism and mineralization, which is very important for young growing organisms, there is more and more evidence confirming the pleiotropic action of vitamin D which could be equally important for children.

The limitation of this study is the lack of direct bone density analysis, which would enrich our knowledge of calcium and phosphate management and its relation to vitamin D levels in patients with IBD. In addition, the results obtained repeatedly, as well as the data from the literature, did not provide unambiguous answers and should continue in larger groups. It has been demonstrated, however, in the pediatric population of IBD patients, that regular monitoring of vitamin D levels in blood and its proper supplementation is required.

## 5. Conclusions

Both children with IBD and the control group reported a vitamin D deficiency which suggests the need for vitamin D supplementation in children with inflammatory bowel disease as well as their healthy peers, especially in adolescents. Although IBD patients are usually under constant medical supervision, more attention should be drawn to maintain an adequate vitamin D level in this group of patients. There are many risk factors of vitamin D deficiency associated with IBD, regardless of disease activity or location, that may lead to serious complications. On the other hand, healthy adolescents are also at risk of developing a vitamin D deficiency and its adverse effects, probably due to the increased metabolic demand at the time of a growth spurt, inappropriate dietary habits or infrequent sun exposure;Daily exposure to sunlight has a beneficial effect on serum vitamin D levels in both study groups. It is, therefore, advisable to promote the presence of children outdoors;Nutritional status, diet and physical activity does not seem to modify the level of vitamin D. However, it has been confirmed that amount of vitamin D consumed from the regular diet in both study groups is too low to cover daily requirements. Moreover, IBD patients have a lower physical activity frequency compared to controls, even with a milder clinical course of the disease. Healthy lifestyle habits should be promoted in all children. All of the above observations require verification in larger study groups.

## Figures and Tables

**Figure 1 nutrients-14-01029-f001:**
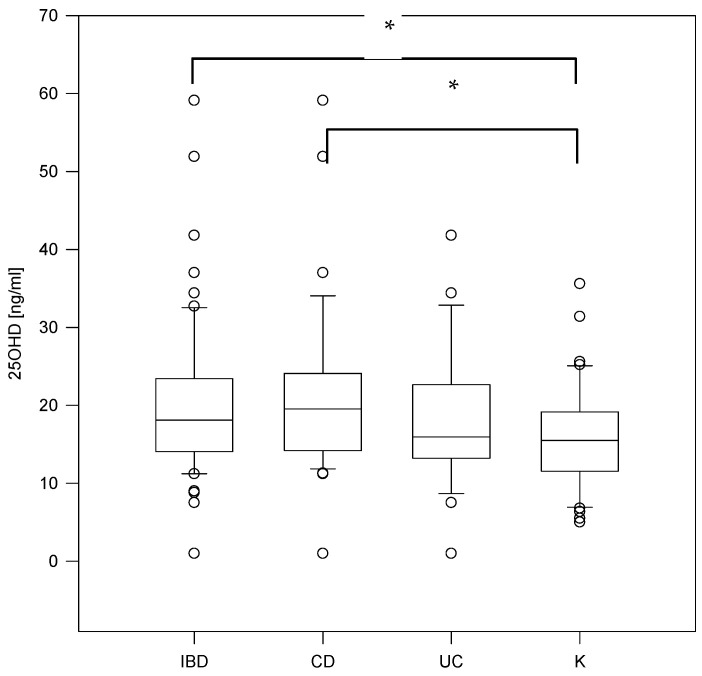
Vitamin D concentration in study groups (* *p* < 0.05; Kruskal–Wallis One Way Analysis of Variance on Ranks test); IBD-Inflammatory Bowel Disease, CD-Crohn’s Disease, UC-Ulcerative Colitis, C-Control/Comparative Group.

**Figure 2 nutrients-14-01029-f002:**
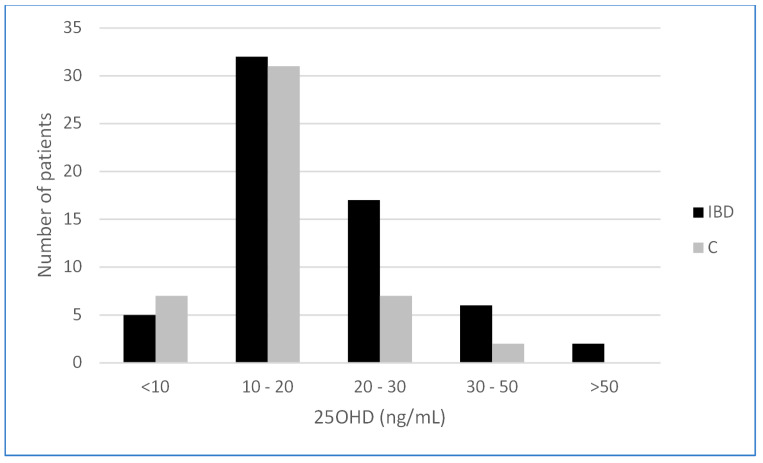
Vitamin D status in study groups (IBD—Inflammatory Bowel Disease, C—control group, pts—patients).

**Table 1 nutrients-14-01029-t001:** Characteristics of study groups.

Characteristic	Number of Patients (%)			
	IBD (*N* = 62)	CD (*N* = 34)	UC (*N* = 28)	C (*N* = 47)
Gender: F M	23 (37) 39 (63)	10 (30) 24 (70)	13 (46) 15 (54)	23 (49) 24 (51)

Season of the 25OHD analysis: summer winter	16 (26) 46 (74)	10 (30) 24 (70)	6 (21) 22 (79)	0 (0) 47 (100)
Clinical course				
PCDAI/PUCAI score: 1/2/3/4		10/16/4/4	13/13/3/0	
Endoscopic location CD: L1/L2/L3/L4 UC: E1/E2/E3/E4		2/21/11/0	5/6/12/5	
Surgical abdominal interventions	5 (8)	4	1	0 (0)
Bone fractures	11 (18)	7	4	3 (6)
Medications:				
Dose of vitamin D supplementation: <600 IU 601–1000 IU >1001 IU	33 (53) 16 (26) 10 (16) 7 (11)	15 8 4 4	18 8 6 3	7 (15) 5 (11) 2 (4) 0 (0)
Steroids	29 (47)	19	10	0 (0)
Steroids + Vitamin D	17 (27)	8	9	0 (0)
Azathioprine	18 (29)	13	5	0 (0)
Biological treatment	12 (19)	9	3	0 (0)
Nutrition: enteral/parenteral	12 (62) 11/1	10/1	0/1	0 (0)

IBD-Inflammatory Bowel Disease, CD-Crohn’s Disease, UC-Ulcerative Colitis, C-Control/Comparative Group, PCDAI-Pediatric Crohn’s Disease Activity Index, PUCAI-Pediatric Ulcerative Colitis Activity Index, IU-International Units.

**Table 2 nutrients-14-01029-t002:** Anthropometric parameters of study groups; ANOVA.

	IBD	CD	UC	C	*p*
Weight	48.5 (37.9–58.0)	48.4 (41.0–53.0)	50.5 (33.5–60.5)	50.0 (42.0–56.77)	NS
Weight z-score	−0.88 ± 1.04	−0.99 ± 1.07	0.74 ± 1.02	−0.014 ± 0.73	*p* < 0.05
Height	1.625 (1.51–1.73)	1.63 (1.57–1.7)	1.62 (1.48–1.75)	1.62 (1.55–1.7)	NS
Height z-score	−0.51 (−1.26–0.29)	−0.51 (−1.03–0.15)	−0.31 (−1.74–0.78)	−0.18 (−0.63–0.29)	NS
BMI	18.16 (16.44–19.53)	17.8 (16.44–19.03)	18.53 (16.43–20.27)	18.52 (17.67–19.72)	NS
BMI z-score	−0.69 ± 1.01	−0.85 ± 0.98	−0.49 ± 1.02	−0.19 ± 0.73	*p* < 0.05

**Table 3 nutrients-14-01029-t003:** Vitamin D level regarding season of the year (summer/winter) and supplementation (supp +/−).

Season/Supplementation	25OHD Level [ng/mL]
IBD summer	14.0 (11.65–26.65)
IBD summer supp +	14.5 (12.38–42.53)
IBD summer supp −	13.4 (11.43–23.48)
IBD winter	19.25 (14.4–23.2)
IBD winter supp +	22.25 (17.85–25.4)
IBD winter supp −	14.45 (13.7–19.2)
C winter	15.5 (11.83–18.7)
C winter supp +	16.2 (12.73–19.38)
C winter supp −	13.55 (9.55–15.85)

Kruskal–Wallis One Way Analysis of Variance on Ranks, IBD winter vs. IBD summer *p* = 0.029.

**Table 4 nutrients-14-01029-t004:** Laboratory blood test results in the study groups.

		Mean Value +/− SD or Median (25–75%)				*p*			
Parameter	Reference Value	CD	UC	IBD	C	CD vs. C	UC vs. C	IBD vs. C	CD vs. UC
25OHD [ng/mL]	(30–50)	19.55 (14.2–23.8)	15.95 (13.3–22.65)	18.1 (14.2–23.3)	15.5 (11.83–18.7)	0.047	NS	0.03	NS
PTH [pg/mL]	(10–62)	22.1 (11.7–31.0)	18.4 (11.25–26.85)	22.0 (11.5–27.9)	17.2 (12.5–24.25)	NS	NS	NS	NS
Ca [mg/dL]	(8.4–10.2)	9.78 (9.3–10.07)	9.78 (9.46–10.06)	9.70 (9.3–10.07)	10.0 (9.81–10.3)	0.002	0.002	<0.001	NS
P [mg/dL]	(2.9–5.1)	4.3 (4.1–4.8)	4.4 (4.0–4.99)	4.35 (4.0–4.9)	4.4 (4.1–5.0)	NS	NS	NS	NS
ALP [U/L]	(<390)	117.5 (95.0–171.0)	124.0 (98.5–200.0)	117.5 (97.0–194.0)	177.0 (90.5–229.5)	NS	NS	NS	NS
Urine Ca/cr ratio [mg/mg]	(0.01–0.25)	0.125 (0.07–0.2)	0.16 (0.09–0.21)	0.13 (0.07–0.2)					NS
ESR [mm/h]		24.5 (13.0–40.0)	25.0 (15.0–35.0)	21.5 (10.5–52.5)					
CRP [mg/L]	(1–5)	2.25 (1.0–15.0)	3.05 (1.0–16.5)	2.75 (1.0–15.0)	1.0 (1.0–1.0)	<0.001	<0.0001	<0.001	NS
WBC [G/L]	(3.5–10)	7.52 (5.4–8.8)	9.18 (7.76–12.05)	8.155 (6.15–10.59)	6.73 (5.73–8.23)	NS	<0.001	0.009	<0.001
Neu [G/L]	(1.8–7.7)	4.235 (2.59–6.5)	5.79 (3.82–8.65)	4.75 (3.02–6.6)	3.86 (3.03–4.75)	NS	<0.001	0.058	NS
Hb [g/dL]	(K:12–15; M:12.5–16.1)	12.14 ± 1.16	12.11 ± 1.77	12.13 ± 1.45	13.62 ± 1.24	<0.001	<0.001	<0.001	NS
PLT [G/L]	(125–400)	351.0 (299.0–420.0)	343.5 (286.5–420.5)	351.0 (290.0–420.0)	268.0 (236.0–324.5)	<0.001	<0.001	<0.001	NS

ANOVA, Kruskal–Wallis One Way Analysis of Variance on Ranks.

**Table 5 nutrients-14-01029-t005:** Frequency of sun exposure of IBD and control group in different times of the year.

Season	Subgroup	Number of Patients in %					*p*
		1	2	3	4	1 + 2	
Spring	IBD	38.71	37.1	17.74	6.45	75.81	0.038
	CD	41.18	35.29	14.71	8.82	76.47	
	UC	35.71	39.29	21.43	35.7	75.00	
	C	31.91	63.83	4.26	0.00	95.74	
Autumn	IBD	29.03	45.16	20.97	4.84	74.19	0.002
	CD	26.47	50.0	17.65	5.88	76.47	
	UC	32.14	39.29	25.0	3.57	71.43	
	C	12.77	85.11	2.13	0.00	97.88	
Summer	IBD	59.68	27.42	8.06	4.84	87.1	0.144
	CD	55.88	29.41	8.82	5.88	85.29	
	UC	64.29	25.0	7.14	3.57	89.29	
	C	40.43	53.19	6.38	0.00	93.62	
Winter	IBD	22.58	41.93	29.03	6.45	64.51	0.0015
	CD	20.59	47.06	23.53	8.82	67.65	
	UC	25.00	35.71	35.71	3.57	60.71	
	C	4.26	85.11	10.64	0.00	89.37	

(1 = everyday, 2 = few times per week, 3 = few times per month, 4 = less than 1× per month).

**Table 6 nutrients-14-01029-t006:** Vitamin D level according to declared frequency of sun exposure during one year in total group of patients (IBD + C).

Points	Vitamin D Level (ng/mL)
0	12.1 (11.43–20.43)
4	14.5 (13.8–19.3)
5	15.75 (13.85–17.9)
6	14.9 (14.2–22.13)
7	8.5 (7.03–20.43)
8	13.4 (11.1–17.1)
9	17.2 (14.0–22.83)
10	20.0 (16.95–28.2)
11	19.6 (17.3–22.53)
12	24.55 (17.6–32.7)

**Table 7 nutrients-14-01029-t007:** Vitamin D level according to declared frequency of sun exposure during the summer in total group of patients (IBD + C).

Points	Vitamin D Level (ng/mL)
0	12.1 (11.43–20.43)
1	13.95 (12.6–15.35)
2	15.5 (11.6–18.7)
3	19.45 (14.0–24.2)

**Table 8 nutrients-14-01029-t008:** Vitamin D level according to declared frequency of sun exposure during the summer in total group of patients (IBD + C).

Points	IBD	C
0–3	12.1 (11.42–20.42)	
4–6	14.5 (14.02–19.6)	14.35 (13.8–14.9)
7–9	15.2 (11.57–22.67)	13.4 (10.85–17.87)
10–12	15.2 (11.57–22.67)	18.75 (16.95–23.5)

IBD *p* = 0.014, C *p* = 0.006.

**Table 9 nutrients-14-01029-t009:** Vitamin D level and frequency of sun exposure in the supplemented (sup) and non-supplemented (no sup) groups (IBD vs. C).

Points	IBS Sup	IBD No Sup	*p*	C Sup	C No Sup	*p*
0–3	23.2 (23.2–23.2)	11.65 (11.2–12.1)	*	*	*	*
4–6	18.9 (14.35–21.25)	14.5 (12.95–15.2)	0.05	13.8 (13.8–13.8)	14.9 (14.9–14.9)	NS
7–9	22.6 (13.65–27.95)	14.2 (11.3–18.1)	0.06	10.7 (7.55–13.42)	13.65 (11.3–18.4)	0.07
10–12	23.8 (18.53–33.98)	19.7 (14.3–24.25)	0.16	26.75 (17.9–35.6)	18.7 (16.2–22.1)	0.19

* due to a small sample size, calculations were not performed.

**Table 10 nutrients-14-01029-t010:** Vitamin D level and frequency of sun exposure (sup) in the supplemented and non-supplemented (n sup) groups (total).

Points	Total Sup	Total No Sup	*p*
0–3	23.2	11.65 (11.2–12.1)	-
4–6	18.5 (14.1–20.87)	14.5 (13.3–14.9)	0.17
7–9	17.14 ± 9.61	14.69 ± 6.07	0.26
10–12	23.8 (18.05–34.7)	19.4 (15.5–22.1)	0.05

**Table 11 nutrients-14-01029-t011:** Frequency of consumption of products rich in vitamin D and calcium among IBD and control group.

		1	2	3	4	
Milk	IBD	22.58	29.03	14.51	33.87	0.0009
	C	34.04	53.19	12.77	0	
Yogurt	IBD	29.03	38.71	16.13	16.13	0.049
	C	34.04	51.06	14.89	0	
Yellow cheese	IBD	38.71	43.55	11.29	6.45	0.02
	C	44.68	55.32	0	0	
Eggs	IBD	3.22	80.64	12.91	3.22	0.00085
	C	0	48.94	51.06	0	
Fat fish	IBD	0	8.06	40.32	51.61	0.00004
	C	0	0	44.68	55.32	

(1 = everyday, 2 = few times per week, 3 = few times per month, 4 = less than 1× per month).

**Table 12 nutrients-14-01029-t012:** Analysis of 3 days’ diary per day in IBD vs. control.

	IBD	C	*p*
Kcal	1650.95 (1483.37–1939.07)	1787.75 (1512.22–2073.3)	NS
%EAR	72.45 ± 14.13	83.4 ± 17.44	<0.001
Protein (g)	69.64 ± 16.07	75.98 ± 16.25	0.044
Protein (g/kg)	1.46 (1.22–1.82)	1.47 (1.27–1.89)	NS
Protein (%)	16.81 (14.31–18.49)	16.02 (14.35–19.17)	NS
Fat (g)	53.85 (45.84–70.02)	56.21 (45.78–83.19)	NS
Fat (%)	30.43 ± 5.76	29.34 ± 47.22	NS
Carbs (g)	232.21 (197.58–273.75)	245.12 (207.99–291.56)	NS
Carbs (%)	52.93 ± 96.14	53.27 ± 7.17	NS
*	15.3:20.5:24.3:17.3:20.8	13.9:20.6:23.6:17.9:20.2	NS
Vit D (IU)	79.63 (56.53–107.56)	85.14 (62.27–114.39	NS
Ca (mg)	587.542 (461.59–758.59)	699.423 (547.32–874.96)	0.034
P (mg)	1051.42 (915.239–1146.65)	1451.02 (1058.46–1305.51)	NS
Ca:P	0.580 (0.49–0.72)	0.62 (0.51–0.72)	NS

* % of calories in 1st breakfast: 2nd breakfast:dinner:snack:supper.

**Table 13 nutrients-14-01029-t013:** Vitamin D level and the type of the therapy (Y—yes, N—no).

Characteristic	25OHD [ng/mL]; Mean ± SD or Median (25–75%)				
	IBD	CD	UC	C	*p*
Vitamin D supplementation	Y–22.0 (16.33–26.08) N–14.4 (12.33–19.33)	Y–23.25 (20.15–28.45) N–14.45 (13.3–19.2)	Y–20.23 ± 9.28 N–15.52 ± 8.16	Y–13.55 (9.55–15.85) N–16.2 (12.73–19.38)	IBD 601–1000 vs. IBD vitD– *p* = 0.002 IBD vitD 601–1000 vs. C vitD- *p* = 0.003 CD 601–1000 vs. CD vitD– *p* = 0.002 CD Y vs. N < 0.001 UC – NS
Dose of vitamin D supplementation:					
<600 IU	19.8 (16.48–22.55)	20.9 (19.55–36.95)	16.68 ± 6.48	13.8 (9.63–16.88)	
601–1000 IU	31.1 (21.18–35.92)	34.05 (25.9–48.05)	23.27 ± 9.12		
>1001 IU	20.5 (15.075–25.2)	23.25 (19.3–24.4)	21.19 ± 10.87	10.7	
Steroids	Y–18.1 (13.38–23.23) N–18.1 (14.2–24.0)	Y–19.75 (14.35–24.15) N–18.95 (14.2–23.8)	Y–15.74 ± 7.71 N–19.63 ± 9.5		NS
Steroids+ vitD+ (S+D+)	20.5 (14.03–23.73)	23.2 (20.2–28.0)	16.2 ± 8.11		IBD D+ S- vs. IBD D- S- *p* = 0.003 IBD D+ S- vs. IBD D- S+ *p* = 0.003 CD D+ S- vs. CD D- S- *p* = 0.004 CD D+ S+ vs. CD D- S- *p* = 0.004
Steroids + vitD– (S+D–)	14.5 (12.85–19.7)	14.5 (13.33–19.95)	12.1 ± 0		
Steroids - vitD+ (S-D+)	22.6 (18.3–29.0)	23.8 (20.35–29.77)	23.81 ± 9.16		
Steroids – vitD– (S–D–)	14.2 (11.75–17.9)	14.2 (11.9–16.43)	15.86 ± 8.52		
Azathioprine	Y–18.1 (13.95–23.28) N–18.1 (14.2–23.53)	*	*		NS
Biological treatment	Y-22.6 (14.0–27.3) N–17.45 (14.2–22.6)	*	*		NS
Nutrition (parenteral/enteral):	Y–19.5 (13.95–30.65) N–17.85 (14.2–23.2)	*	*		NS

* due to a small sample size, calculations were not performed. ANOVA, Kruskal–Wallis One Way Analysis of Variance on Ranks and One Way Analysis of Variance.

## Data Availability

Data are available upon request.

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
