# Peer review of "Diet, Sun, Physical Activity and Vitamin D Status in Children with Inflammatory Bowel Disease"

_nutrients, 2022, doi:10.3390/nu14051029_

Round 1
Reviewer 1 Report
Following the analysis of the article "Diet, sun, physical activity and vitamin D status in children with inflammatory bowel disease" are some aspects that should be specified by the author, respectively
- The novelty of the study should be clearly stated in the discussion section
- Regarding the frequency consumption of calcium and vitamin D rich food the questionnaire used is it part of a validated questionnaire? This important aspect should be specified for the validity of the results
- The conclusions are too general and do not reflect the novelty brought by the study. It needs to be reformulated.
Author Response
Reviewer 1
R: Authors would like to thank the Editor and Reviewers for kind and very helpful evaluation of our article. All answers to the queries are placed within the original review and the manuscript was amended accordingly.
Following the analysis of the article "Diet, sun, physical activity and vitamin D status in children with inflammatory bowel disease" are some aspects that should be specified by the author, respectively
- The novelty of the study should be clearly stated in the discussion section
R: It has been stated in lines:277-278:” To our knowledge, this is the first study that considers multiple variables that may influence the vitamin D level in IBD children” In an addition our novel, complex approach was also underline in the abstract “Here, we are presenting complex analyses of multiple factors influencing 25OHD levels …” as well as at the end of introduction: “. This represents novel and complex analyses of factors affecting vitamin D status in children with IBD.”
- Regarding the frequency consumption of calcium and vitamin D rich food the questionnaire used is it part of a validated questionnaire? This important aspect should be specified for the validity of the results
R: The questionnaire was formed on a basis of FFQ-6 (Food Frequency Questionnaire), but as we were interested mostly in the consumption of vitamin D and calcium - rich products, we adapted the form to our specific needs. To receive more detailed information, we performed further analysis of 3-days dietary diary, with program DIETA 5.
- The conclusions are too general and do not reflect the novelty brought by the study. It needs to be reformulated.
Thank you for your advice, I have made all of the suggested changes.
Reviewer 2 Report
The paper describes vitamin D levels in children with inflammatory bowel disease in comparison to those without.
I think some of the results were over interpreted eg the increased number of fractures. There was no significant difference for many outcomes, even though there may have been a trend towards a difference. These should not be shown in tables in the text or discussed with such prominence and no claims should be made about them. There is a statement in the discussion that there were few fractures recorded. A small amount of discussion would be appropriate as it is an important part of the literature – more work is needed to increase the sample size to be able to make definitive conclusions, something not possible here. This would decrease the number of tables in the results. The data could go into supplementary material so that if a meta analysis is ever done, this could contribute. This also applies to exercise and sun exposure.
The descriptions of questions asked (sun exposure, physical activity, diet) and how the scores were calculated should be in the methods not the results section.
It’s not very clear when the blood samples were taken or how many – this should be in the methods section. Table 3 suggests several samples throughout the year, line 357 suggests they were mainly taken in winter.
English: Introduction is written as a single paragraph. Breaking this down into 3-5 paragraphs would improve readability. Section 4.2 could also be put into paragraphs
In the discussion there is the consistent use of the word ‘what’ in the discussion eg line 443 ‘what explained’. This should be ‘which’ – ‘which explained’. Other lines noted 429, 460, 475, 481, 492
Author Response
Reviewer 2
R: Authors would like to thank the Editor and Reviewers for kind and very helpful evaluation of our article. All answers to the queries are placed within the original review and the manuscript was amended accordingly. All changes to the manuscript were marked in the text using track changes feature.
The paper describes vitamin D levels in children with inflammatory bowel disease in comparison to those without.
I think some of the results were over interpreted eg the increased number of fractures. There was no significant difference for many outcomes, even though there may have been a trend towards a difference. These should not be shown in tables in the text or discussed with such prominence and no claims should be made about them. There is a statement in the discussion that there were few fractures recorded. A small amount of discussion would be appropriate as it is an important part of the literature – more work is needed to increase the sample size to be able to make definitive conclusions, something not possible here. This would decrease the number of tables in the results. The data could go into supplementary material so that if a meta analysis is ever done, this could contribute. This also applies to exercise and sun exposure.
The descriptions of questions asked (sun exposure, physical activity, diet) and how the scores were calculated should be in the methods not the results section.
Thank you for your advice, We made all of the suggested changes. Please, see the manuscript.
It’s not very clear when the blood samples were taken or how many – this should be in the methods section. Table 3 suggests several samples throughout the year, line 357 suggests they were mainly taken in winter.
English: Introduction is written as a single paragraph. Breaking this down into 3-5 paragraphs would improve readability. Section 4.2 could also be put into paragraphs
Thank you for your advice, I made all of the suggested changes.
In the discussion there is the consistent use of the word ‘what’ in the discussion eg line 443 ‘what explained’. This should be ‘which’ – ‘which explained’. Other lines noted 429, 460, 475, 481, 492
Thank you for your advice, I made all of the suggested changes.
I an addition the manuscript was carefully read and corrected.